# Possible Association between Bladder Wall Morphological Changes on Computed Tomography and Bladder-Centered Interstitial Cystitis/Bladder Pain Syndrome

**DOI:** 10.3390/biomedicines9101306

**Published:** 2021-09-24

**Authors:** Jia-Fong Jhang, Yung-Hsiang Hsu, Han-Chen Ho, Yuan-Hong Jiang, Cheng-Ling Lee, Wan-Ru Yu, Hann-Chorng Kuo

**Affiliations:** 1Department of Urology, Hualien Tzu Chi Hospital, Buddhist Tzu Chi Medical Foundation, Tzu Chi University, Hualien 970, Taiwan; alur1984@hotmail.com (J.-F.J.); redeemerhd@gmail.com (Y.-H.J.); leecl@hotmail.com (C.-L.L.); 2Department of Pathology, Hualien Tzu Chi Hospital, Buddhist Tzu Chi Medical Foundation, Tzu Chi University, Hualien 970, Taiwan; yhhsu@mail.tcu.edu.tw; 3Department of Anatomy, Tzu Chi University, Hualien 970, Taiwan; hcho@gms.tcu.edu.tw; 4Department of Nursing, Hualien Tzu Chi Hospital, Buddhist Tzu Chi Medical Foundation, Hualien 970, Taiwan; wanzu666@gmail.com

**Keywords:** bladder, histopathology, chronic inflammation, cystitis

## Abstract

This study aimed to evaluate the clinical significance of urinary bladder wall thickening on computed tomography (CT) among patients with interstitial cystitis/bladder pain syndrome (IC/BPS). Patients with IC/BPS were prospectively enrolled and classified into three groups according to bladder CT finding: smooth bladder wall, focal bladder thickening, and diffuse bladder thickening. Among the 100 patients with IC/BPS, 49, 36, and 15 had smooth bladder wall, focal bladder thickening, and diffuse bladder thickening on CT, respectively. Patients with Hunner’s lesion showed a higher proportion of diffuse and focal bladder thickening compared to those without the same (*p* < 0.001). Patients with diffuse bladder thickening displayed smaller first sensation of filling, cystometric bladder capacity, and voided volume compared to the rest (all *p* < 0.001). Patients with focal and diffuse thickening had a higher proportion of inflammatory cell infiltration, uroepithelial cell denudation, and granulation tissue compared to those with smooth bladder wall (*p* = 0.045, 0.002, and 0.005, respectively). Bladder wall thickening on CT was correlated with clinical phenotypes of IC/BPS, including histopathological findings. Focal or diffuse bladder wall thickening on CT might indicate the presence of chronic bladder wall inflammation and fibrosis and could be used to differentiate bladder-centered IC/BPS.

## 1. Introduction

IC/BPS is a highly heterogeneous symptom syndrome characterized by urinary frequency, urgency, and bladder pain [1]. Prominent pathological findings of bladder biopsies in patients with IC/BPS include urothelial denudation and increased inflammatory cell infiltration [2,3]. The bladder wall inflammation is more prominent in the HIC patients compared to the NHIC patients [2,3]. Recently, our group found a significant correlation between MBC and grade of glomerulation hemorrhage, with both being significantly associated with bladder inflammation and ICSI [2,4]. Despite the highly heterogeneous pathogenesis of IC/BPS, the severity of bladder wall inflammation likely affects the severity of symptoms, grade of glomerulation, and MBC in the patients with bladder-centered IC/BPS.

Apart from lower urinary tract symptoms, a considerable number of patients with IC/BPS present with a functional somatic syndrome and multiple somatic pains and functional disorders [5]. Recently, non-bladder-centered IC/BPS has been proposed to characterize for a group of patients presenting with affect dysregulation and somatic syndrome [6]. Accordingly, the International Society for the Study of Interstitial Cystitis/Bladder Pain Syndrome (ESSIC) classified IC/BPS subtypes according to the bladder histopathological and cystoscopic findings [7]. Our previous studies in a large cohort of patients with IC/BPS revealed that approximately 41.6% and 21.2% of patients were ESSIC type 1 (grade of glomerulation hemorrhage ≤1) and ESSIC type A (no abnormal histopathology findings) [2,8]. Patients with IC/BPS who had no significant cystoscopic or histopathology finding are classified into ESSIC type 1A, which suggests that the symptoms in these patients might be caused by extrabladder disorders [7,9]. Evidence has shown that differentiation from IC/BPS to bladder-centered or extrabladder disorder have potential to improve treatment efficacy [2]. However, given inconvenience and invasiveness of using cystoscopy and biopsy to classify bladder-centered IC/BPS, urologists require a more practical and less invasive tool to differentiate IC/BPS subtypes.

Considering that the guidelines recommend partial cystectomy and bladder augmentation for patients with HIC refractory to conservative treatment [1,9], we had previously performed this procedure for patients with HIC [10], with bladder CT being routinely performed before surgery. Accordingly, our group had noted that all patients with HIC exhibited diffuse or focal bladder wall thickening on bladder CT. As such, it may be rational to assume correlations between bladder wall inflammation and bladder wall thickness, histopathological findings, and clinical presentations of IC/BPS. The current study therefore aimed to investigate whether bladder wall thickening on CT could be observed in patients with both HIC and NHIC, as well as explore the possible role of bladder wall thickness on CT as a predictor of bladder-centered IC/BPS.

## 2. Materials and Methods

A total of 100 consecutive patients with confirmed IC/BPS were enrolled herein. All patients had been diagnosed with IC/BPS using the ESSIC criteria [7]. Patients who had undergone any urological procedure over the recent 6 months were excluded. Detailed patient inclusion and exclusion criteria are listed in Appendix A. This study was approved by the Ethics Committee (IRB: 105-25-B), and informed consent was obtained from the patients prior to participation. Symptom scores, including ICSI and ICPI, OSS (=ICP + ICSI), VAS, and GRA for previous treatments were recorded. All patients underwent videourodynamic examination to rule out BOO and the other differential diagnoses. Cystoscopic HD was routinely performed, with patients receiving intravesical treatment, including onabotulinumtoxin A injection or electrocauterization for Hunner’s lesions. To investigate bladder morphology in patients without urologic problem, 20 sex- and age-matched patients in the general surgery and cardiovascular surgery wards who had undergone abdominal CT and had no history of urologic diseases were also retrospectively enrolled as control subjects.

Pelvis bladder CT was performed upon recent admission to evaluate bladder wall thickness and other possible pathologies. Bladder volume was kept at 50 to 100 mL given that most patients with IC/BPS could not tolerate a full bladder. Bladder CT findings were classified as smooth bladder wall when was even thickness throughout the bladder was observed, focal thickening when bladder wall thickening was observed in only a portion of bladder, or diffuse bladder wall thickening when bladder wall thickening involved over half of the bladder. (Figure 1) All bladder CT groupings were performed by single urologist (HC Kuo) who was not involved in the final data analysis.

Patients then underwent cystoscopic HD and bladder biopsy under general anesthesia. The bladder biopsy procedure was identical to that in our previous study [2]. The MBC and grading of glomerulation hemorrhage after HD were recorded. Histological findings of bladder biopsies, including inflammatory cell infiltration, uroepithelial cell denudation, fibrosis, eosinophil infiltration, plasma cell infiltration, lamina propria hemorrhage, or presence of granulation tissue, were graded by a pathologist (YH Hsu), similar to our previous study [2]. Histopathological grading definitions are presented in Appendix A. Patients with a least one positive histopathological finding were classified as ESSIC type C, while those without any positive histopathological findings were classified as ESSIC type A [2,7]. Moreover, five patients in each CT group underwent Masson’s trichrome staining to determine collagen deposition in the bladder specimens.

### Statistical Analysis

Patients were classified into three groups according to bladder wall thickness. Patient demographics, clinical symptoms, urodynamic parameters, cystoscopic features, and histopathology findings between the three groups were then compared. Differences in symptom scores and objective parameters between the groups were compared using ANOVA. The Chi-square test was used to analyze non-continuous variables, including IC/BPS subtypes distribution, grading of glomerulation hemorrhage, and histopathological findings. All statistical analyses were performed using SPSS for Windows, version 20.0 (SPSS, Chicago, IL, USA), with *p*-values < 0.05 indicating statistical significance.

## 3. Results

According to bladder CT classification, 49, 36, and 15 patients with IC/BPS presented with a smooth bladder wall, focal thickening, and diffuse thickening, respectively. Among the control patients (17 female and 3 male), only 1 male patient (5%) had diffuse bladder wall thickening, while the rest had smooth bladder wall. No difference in age, disease duration, and gender distribution were observed between the CT groups (Table 1). A total of 36 patients with IC/BPS had a history of abdominal or pelvic surgery, with no significant difference in their proportion within the three groups (Table 1; *p* = 0.349). The bladder CT findings of bladder wall thickening could correspond to cystoscopic HD findings (Figure 2). Accordingly, patients with HIC had a higher proportion of bladder diffuse (86.7%) and focal (30.6%) thickening compared to those with NHIC (Table 1; *p* < 0.0001). Interestingly, patients with NHIC also exhibited focal (32.9%) and even diffuse (2.6%) bladder wall thickening. Although ICSI tended to significantly increase from smooth bladder wall to focal and then diffuse thickening, no significant difference in the ICPI, OSS, VAS, and treatment outcomes measured using GRA were observed between the groups (Table 1). Notably, significant differences in objective urodynamic parameters were observed between the groups. Patients with diffuse bladder thickening had smaller first sensation of filling, fullness sensation, cystometric bladder capacity, and voided volume compared to the rest of the groups (all *p* < 0.001), although no significant difference in Pdet was noted. After excluding patients with HIC, patients with diffuse bladder thickening still showed smaller CBC (Table 1).

During cystoscopic HD, patients with a smooth bladder wall had the largest MBC, followed by those with focal thickening and those with bladder diffuse thickening, who had smallest MBC (Table 1; *p* < 0.0001). Differences remained significant even after excluding patients with HIC. Patients with HIC/grade 4 glomerulation had more focal or diffuse bladder wall thickening than those with grade 0–3 glomerulation NHIC (Figure 3A; *p* < 0.001). The proportion of bladder thickness was not significantly different among the NHIC patients. According to ESSIC cystoscopic classification, patients with ESSIC type 3 IC/BPS had more focal or diffuse bladder wall thickening (Figure 3B, *p* < 0.001), although no significant difference was observed between ESSIC type 1 and type 2. After further classifying patients according to the combination of MBC and glomerulation in patients with NHIC, our results showed that those with small MBC (<760 mL) and grade of glomerulation 0–1 had significantly higher proportions of focal or diffuse bladder wall thickening (Figure 3B; *p* = 0.003).

The association between bladder wall thickness on CT and histopathology results are detailed in Table 2. Accordingly, patients with diffuse bladder wall thickening had the greatest inflammatory cell infiltration, uroepithelial cell denudation, and granulation tissue, followed by those with focal thickening (Table 2). Patients with focal or diffuse bladder thickening had a higher proportion of ESSIC type C than ESSIC type A. Although no significant difference in fibrosis was observed under H&E staining, Masson’s trichrome staining revealed differences in histological fibrosis characteristics between the CT groups. Accordingly, patients with smooth bladder wall showed no obvious collagen accumulation in the bladder specimens (Figure 4A–C), with only a few fine collagen fibers having been found. In contrast, those with focal bladder thickening had obvious collagen deposition in the bladder deep lamina propria (Figure 4D–F), while those with diffuse bladder thickening even had thick collagen fiber accumulation in both superficial and deep lamina propria (Figure 4G–I).

## 4. Discussion

Since the discovery of Huuner’s lesion more than 100 years ago, patients with IC/BPS had generally underwent cystoscopy, urodynamic study, and biopsy for histopathology, with radiological examinations for gross bladder morphology being rare. Several recent pilot studies, however, have utilized MRI to evaluate bladder wall changes in patients with IC/BPS [11,12,13]. Although evidence had suggested that MRI could be used for noninvasive diagnosis of IC/BPS, the association between clinical characteristics and abnormal findings have yet to be analyzed. After prospectively enrolling a large cohort of patients with IC/BPS, the current study revealed that bladder wall thickness on CT was associated with the clinical phenotypes of IC/BPS. Moreover, our findings showed that focal and diffuse bladder wall thickening had been associated with smaller MBC, higher grade of glomerulation, presence of Hunner’s lesion, histopathology finding, urodynamic storage parameters, and clinical symptoms. Changes in bladder wall morphology on CT could therefore have the potential to differentiate bladder-centered disease among patients with IC/BPS.

Bladder wall thickening is not a specific finding on imaging studies and might be caused by BOO, bladder cancer, inflammation, previous abdominal, or pelvic operation [14]. The current study found that more than half of the patients with IC/BPS had bladder wall thickening, whereas only 5% of control subjects revealed the same. Despite excluding patients with BOO from urodynamic study, those with IC/BPS who had bladder wall thickening did not show higher Pdet, suggesting that the findings could not be attributed to BOO. Histopathological results revealed that bladder wall thickening on CT was associated with urothelium denudation, inflammatory cell infiltration, and presence of granulation tissue. Moreover, Masson’s trichrome staining showed that patients with bladder wall thickening exhibited significant collagen deposition and fibrosis. Our results suggest that bladder wall thickening on CT might indicate focal or diffuse bladder fibrosis, which could be attributed to chronic inflammation. CT bladder wall thickening indeed represented bladder tissue pathological changes.

The taxonomy of IC/BPS has long been deliberated with no conclusion having yet been established. However, recent investigations have gradually identified which patients with IC/BPS should be categorized under HIC and NHIC [15]. Although patients with HIC might have more severe bladder symptoms and small bladder capacity, identifying those with HIC from patients with IC/BPS without cystoscopy remains challenging [9,15]. Even under cystoscopy, the diagnosis of HIC has still been contentious, with the prevalence of HIC reported to range from 5 to 57% through different regions of the world [16]. With the aid of a Hunner’s lesion atlas, urologists could improve their consensus regarding the diagnosing of HIC through office cystoscopy [17]. However, intra-observer reliability in the identification of characteristic Hunner’s lesion has still been somewhat controversial. The current study revealed that all patients with HIC presented with focal or diffuse bladder wall thickening on CT. Thus, bladder CT should be a more convenient and less invasive tool for differentiating IC/BPS, although cystoscopy still remains necessary to confirm the diagnosis of HIC.

Current guidelines suggest that patients with HIC should receive more aggressive treatments, such as triamcinolone injection, transurethral electrocauterization/resection, and even partial cystectomy for Hunner’s lesions [1,7,9]. However, Hunner’s lesion recurrence has been common and unpredictable, which might involve both previous transurethral resection and cauterization sites and de novo areas [18]. The inflammatory areas in HIC could very likely involve focal or even the entire bladder wall, with the most inflamed lesions possibly erupting and occasionally presenting as Hunner’s lesions [19]. The other inflammatory tissues might remain in the HIC bladder wall and lead to symptom relapse after surgical intervention. However, despite video-urodynamic study during cystoscopy, areas presenting inflammatory changes in the HIC bladder wall still cannot be clearly identified. Imaging studies, such as such as CT or MRI, could provide a better alternative for visualizing changes in the entire bladder wall morphology among patients with IC/BPS. Bladder imaging studies, such as CT, could allow the mapping of bladder lesions, which would enable urologist to attempt at addressing all inflammatory tissues through endoscopic resection/eletrocauterization or partial cystectomy for Hunner’s lesions all at once. For patients with NHIC, focal bladder wall thickening also suggests the chronic inflammation might be present in the involved bladder wall. Treatment of bladder focal thickening through intravesical injection or cauterization among patients with NHIC could also be a reasonable option. Treatment focused on the thick bladder wall portion might effectively eradicate chronic inflammation and improve IC/BPS symptoms.

Recently, investigators have proposed that patients with IC/BPS who had significant dysregulation and somatic syndrome could be identified as a distinct group and could be charactered as non-bladder-centered IC/BPS [6]. patients with IC/BPS who had large MBC and grade 0–1 glomerulation had more systemic comorbidities compared to other IC/BPS cases, suggesting a non-bladder-centered feature in this group [4]. Our recent study also revealed differences in urinary inflammatory cytokine expression between ESSIC type 1 and 2 IC/BPS, similarly suggesting the non-bladder-centered feature in ESSIC type 1 [20]. For patients with IC/BPS who had no positive bladder histopathology finding (ESSIC type A), treatments only focused on bladder might lead to less effective outcomes [2]. By combining cystoscopy glomerulation, MBC, urinary cytokines expression, histopathology, and bladder CT results, urologists could more clearly differentiate non-bladder-centered IC/BPS. Furthermore, individualized treatments for non-bladder-centered IC/BPS might provide better outcomes.

The main limitations of the current study are lack of standardization for bladder volume during CT given that patients with IC/BPS could not hold much urine. Hence, bladder wall thickening could be not precisely determined through a quantified definition. However, the generally classification used herein is convenient and suitable for clinical practice. Moreover, treatment outcomes were only assessed by but not an objective parameter. Although patients who underwent intravesical treatment over the recent 6 months had been excluded, most patients had previously received different intravesical treatment and bladder biopsy, which might have cause focal inflammation and scarring in the bladder.

## 5. Conclusions

The current study revealed that bladder wall thickening on CT was correlated with the clinical phenotypes of IC/BPS, as well as histopathological findings. Focal or diffuse bladder wall thickness on CT might therefore indicate the presence of chronic bladder wall inflammation and fibrosis and is commonly found in patients with IC/BPS who had decreased MBC, increased glomerulation, or Hunner’s lesion. Taken together, bladder wall thickening on CT could be used to differentiate bladder-centered IC/BPS.

## Figures and Tables

**Figure 1 biomedicines-09-01306-f001:**
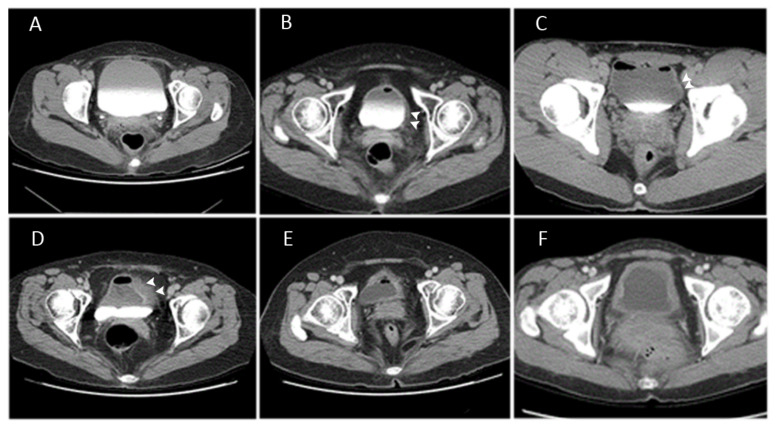
Representative computed tomography (CT) scan bladder images in patients with interstitial cystitis/bladder pain syndrome: (**A**) smooth bladder wall; (**B**–**D**) focal bladder wall thickening (white arrow head); (**E**,**F**) diffuse bladder wall thickening.

**Figure 2 biomedicines-09-01306-f002:**
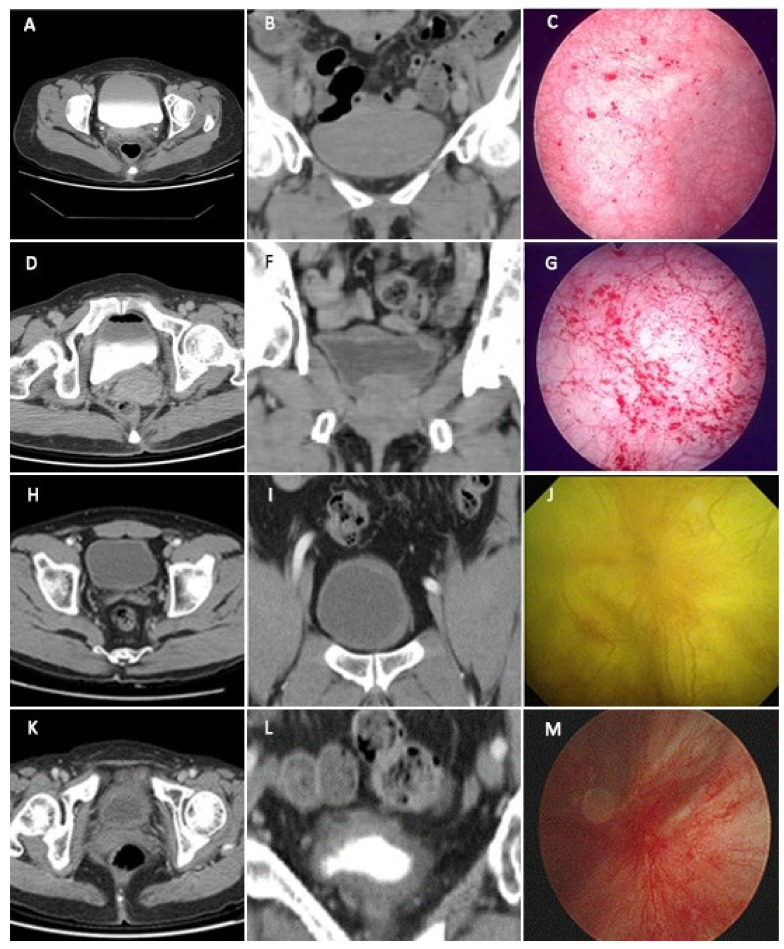
Bladder wall thickness and cystoscopic HD features in patients with interstitial cystitis/bladder pain syndrome. (**A**–**C**) Patient with smooth bladder wall and grade 1 glomerulation; (**D**–**G**) patient with focal thickening bladder wall and grade 2 glomerulation, (**H**–**J**) patient with focal thickening bladder wall and Hunner’s lesion; (**K**–**M**) patient with diffuse thickening bladder wall and Hunner’s lesion.

**Figure 3 biomedicines-09-01306-f003:**
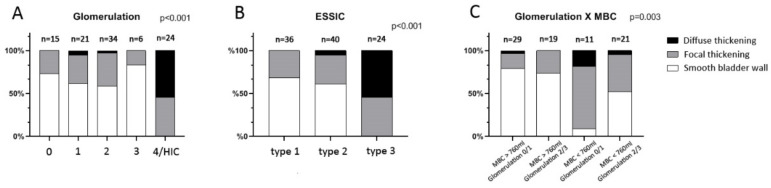
Association between various interstitial cystitis/bladder pain syndrome classification and bladder wall thickening on computed tomography: (**A**) subgrouping according to grade of glomerulation hemorrhage; (**B**) subgrouping according to ESSIC cystoscopy classification; (**C**) subgrouping according to combination of MBC and grade of glomerulation.

**Figure 4 biomedicines-09-01306-f004:**
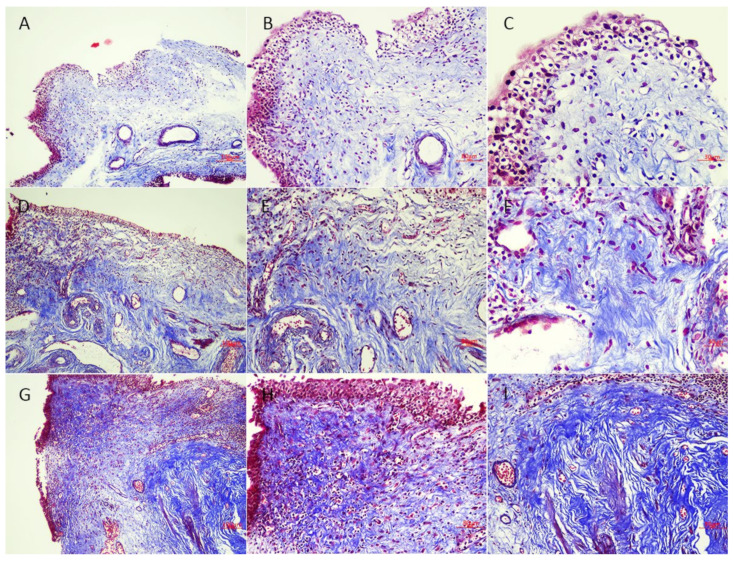
Bladder Masson’s trichrome staining characteristics in each computed tomography (CT) groups. (**A**–**C**) Patients with smooth bladder wall on CT. The specimens showed no obvious collagen accumulation in the bladder, only a few fine collagen fibers could be found. (**D**–**F**) The patients with focal bladder thickening. Obvious collagen deposition in the deep lamina propria was noted. (**G**–**I**) Patients with diffuse bladder thickening. Thick collagen fiber accumulated in both superficial and deep lamina propria. Scale bar = 100 μm (**A**,**D**,**G**), 80 μm (**B**,**E**,**H**,**I**), and 30 μm (**C**,**F**).

**Table 1 biomedicines-09-01306-t001:** Patient demographics and clinical data of the 100 patients with interstitial cystitis/bladder pain syndrome.

	(A) Smooth Bladder Wall (n = 49)	(B) Focal Thickening (n = 36)	(C) Diffuse Thickening (n = 15)	*p* Value (Significant in Post Hoc Analysis)	*p*-Value Excluding HIC(Significant in Post Hoc Analysis)
Age (years old)	53.2 ± 14.5	57.4 ± 10.8	52.1 ± 12.2	0.251	0.771
Gender	45 F, 4 M	28 F, 8 M	13 F, 2 M	0.181	0.139
Abdominal-pelvic surgery history	21 (42.9%)	10 (27.8%)	5 (33.3%)	0.349	
Duration (year)	11.8 ± 8.8	13 ± 10.1	8.7 ± 6.6	0.313	0.234
HIC	0	11 (30.6%)	13 (86.7%)	<0.0001	
NHIC	49 (64.5%)	25 (32.9%)	2 (2.6%)		
ICSI	11.9 ± 3.9	12.9 ± 3.9	16.7 ± 3.4	<0.001 (A v B; B v C)	0.410
ICPI	12 ± 3.5	12.7 ± 2.8	14.2 ± 3.4	0.080	0.411
OSS	23.4 ± 7.8	24.9 ± 7.5	26.8 ± 12.5	0.378	0.496
VAS	5.4 ± 2.6	5.5 ± 3.4	6.5 ± 3.4	0.448	0.346
GRA	1.2 ± 1.4	1.2 ± 1.0	2.0 ± 0.9	0.245	
FSF (mL)	131.7 ± 51.3	125.9 ± 43.2	73.3 ± 32.9	<0.001 (A v C; B v C)	0.734
FS (mL)	208 ± 76	185.4 ± 58.3	100.9 ± 46.8	<0.001A v C; B v C	0.563
CBC (mL)	271.6 ± 111.8	249.5 ± 87.6	135.0 ± 76.1	<0.001 (A v C; B v C)	0.040 (A V C)
Compliance	63.9 ± 54.3	55.9 ± 37.8	34.5 ± 24.3	0.095	0.782
Pdet (cm H_2_O)	17.8 ± 8.7	19.1 ± 11	17.8 ± 8.7		
Qmax (mL/s)	10.8 ± 5.0	10.7 ± 5.7	7.9 ± 4	0.157	0.205
Voided volume (mL)	237 ± 114	189 ± 96.7	111 ± 76.6	<0.001 (A v C; B v C)	0.081
PVR (mL)	39.0 ± 77.1	66.0 ± 114	26.6 ± 38.4	0.249	0.920
MBC (mL)	838.8 ± 182.1	663.1 ± 178.7	392.7 ± 182.0	<0.001 (A v C; B v C; A v B)	0.006 (Av B)

ICSI: IC symptom index, ICPI: IC problem index, OSS: O’Leary Sant symptom score, VAS: visual analog scale of pain, FSF: first sensation of filling, FS: full sensation, CBC: cystometric bladder capacity, Qmax: maximum flow rate, Pdet: detrusor pressure at Qmax, PVR: post-void residual volume.

**Table 2 biomedicines-09-01306-t002:** Bladder wall thickness and histopathological findings in patients with interstitial cystitis/bladder pain syndrome.

		(A) Smooth Bladder Wall (n = 49)	(B) Focal Thickening (n = 36)	(C) Diffuse Thickening (n = 15)	*p*-Value
Inflammatory cells infiltration	None	12 (25.4%)	3 (8.3%)	0	0.045
	Mild	28 (57.1%)	24 (66.7%)	10 (66.7%)	
	Moderate	9 (18.4%)	9 (25.0%)	4 (33.3%)	
	Severe	0	0	1 (6.7%)	
Uroepithelial cells denudation	None	29 (59.2%)	15 (41.7%)	2 (13.3%)	0.002
	Mild	17 (34.7%)	13 (36.1%)	5 (33.3%)	
	Moderate	3 (6.1%)	4 (11.1%)	5 (33.3%)	
	Severe	0	4 (11.1%)	3 (20%)	
Fibrosis	Present	18 (36.7%)	10 (27.8%)	5 (33.3%)	0.686
	Non-Present	31 (63.3%)	26 (72.2%)	10 (66.7%)	
Plasma cell infiltration	Present	8 (16.3%)	11 (30.6%)	5 (33.3%)	0.207
	Non-Present	41 (83.7%)	24 (69.4%)	10 (66.7%)	
Eosinophil infiltration	Present	6 (12.2%)	7 (19.4%)	6 (40%)	0.056
	Non-Present	43 (87.8%)	29 (80.6%)	9 (60%)	
Hemorrhage of lamina propria	Present	2 (4.1%)	3 (8.3%)	0	0.423
	Non-Present	47 (95.9%)	33 (91.7%)	15 (100%)	
Granulation tissue	Present	4 (8.2%)	7 (19.4%)	7 (46.7%)	0.005
	Non-Present	45 (91.8%)	29 (80.6%)	8 (53.3%)	
ESSIC classification	Type A Type C	12 (24.5%) 37 (75.5%)	2 (5.6%) 34 (94.4)	0 15 (100%)	0.011

## Data Availability

Data are available if contact the corresponding authors.

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
