# Peer review of "Possible Association between Bladder Wall Morphological Changes on Computed Tomography and Bladder-Centered Interstitial Cystitis/Bladder Pain Syndrome"

_biomedicines, 2021, doi:10.3390/biomedicines9101306_

Round 1
Reviewer 1 Report
This study investigated the association of bladder wall CT morphology and clinicopathological findings in patients with IC/BPS. The authors demonstrated that bladder wall thickness (focal/diffuse) was associated with cystoscopic findings, cystometry parameters, and degree of histological inflammation, and concluded that bladder CT morphology could predict the presence of bladder-centric pathology.
The aim and results of this study were very interesting. I agree that IC/BPS categorization should be clearly recognized. At present, the only way to distinguish HIC and BPS is cystoscopy, but it is a kind of invasive test. Thus, invention of less invasive test that clearly and accurately categorize IC/BPS phenotypes is an urgent need. In this regard, the present study is of significance.
However, there are several concerns to be resolved before consideration for publication.
- In figure 3A, after excluding HIC group data, glomerulation grades do not seem to associated with bladder wall thickness. How about the results excepting HIC in association between Glo grades and bladder wall thickness?
- Likewise in figure 3C, MBC less than 760mL with Glo grade 0-1 (ESSIC 1) had higher proportion of bladder wall thickness than Glo grade 2-3 (ESSIC 2). I figured out that the authors intended to demonstrate that bladder wall thickness represent the degree of bladder wall inflammation and its-related etiology, right? If so, why not higher Glo grades groups had higher proportion of bladder wall thickening.
- Images of figure 4 (DEF, GHI) are supposed to be from HIC cases. How about the images of non-Hunner lesion subtypes with diffuse thickness (n = 3). Are they as histologically changed as those images of D-F, or G-I? Please present the images of focal and diffuse bladder wall thickened cases of both HIC and non-Hunner lesion IC, if possible.
- The bladder wall of patients with end-stage HIC eventually get thinner. Thus I guess the manuscript title should be changed as "Possible Association Between Bladder Wall Morphological Changes on Computed Tomography and Bladder-Centered Interstitial Cystitis/Bladder Pain Syndrome"
Author Response
This study investigated the association of bladder wall CT morphology and clinicopathological findings in patients with IC/BPS. The authors demonstrated that bladder wall thickness (focal/diffuse) was associated with cystoscopic findings, cystometry parameters, and degree of histological inflammation, and concluded that bladder CT morphology could predict the presence of bladder-centric pathology.
The aim and results of this study were very interesting. I agree that IC/BPS categorization should be clearly recognized. At present, the only way to distinguish HIC and BPS is cystoscopy, but it is a kind of invasive test. Thus, invention of less invasive test that clearly and accurately categorize IC/BPS phenotypes is an urgent need. In this regard, the present study is of significance.
However, there are several concerns to be resolved before consideration for publication.
- In figure 3A, after excluding HIC group data, glomerulation grades do not seem to associated with bladder wall thickness. How about the results excepting HIC in association between Glo grades and bladder wall thickness?
Reply: Thanks for your comment. The proportion of bladder wall thickness was not significantly different among the NHIC patients. We had added this into the result section. (page 4, line 37 to page 5, line 1)
- Likewise in figure 3C, MBC less than 760mL with Glo grade 0-1 (ESSIC 1) had higher proportion of bladder wall thickness than Glo grade 2-3 (ESSIC 2). I figured out that the authors intended to demonstrate that bladder wall thickness represent the degree of bladder wall inflammation and its-related etiology, right? If so, why not higher Glo grades groups had higher proportion of bladder wall thickening.
Reply: Thanks for your comment. Although bladder glomerulation hemorrhage has been widely investigated in the patients with IC/BPS, the clinical significance and association of histopathology findings are still controversial. Our previous study revealed the bladder glomerulation was not significantly associated with bladder inflammatory cells infiltration (J Urol. 2021;205(1):226-235, PMID: 32856961). Glomerulation might be indicated for certain bladder abnormal change (such as suburothelial granulation in our previous study), but the underlying meaning for the glomerulation still unclear. Current study suggested bladder wall thickness in CT was associated with bladder abnormal histopathology findings (including inflammatory cells infiltration), but the association between the glomerulation was not significant in the NHIC patients.
- Images of figure 4 (DEF, GHI) are supposed to be from HIC cases. How about the images of non-Hunner lesion subtypes with diffuse thickness (n = 3). Are they as histologically changed as those images of D-F, or G-I? Please present the images of focal and diffuse bladder wall thickened cases of both HIC and non-Hunner lesion IC, if possible.
Reply: Thanks for your comment. Indeed, it is interesting to see if bladder fibrosis also presented in the NHIC patients with bladder wall thickness. We had reviewed the specimens from the patients with NHIC and bladder wall thickness, however, the specimens were not large enough to perform Masson's trichrome staining to see if collagen deposition presented in the submucosal tissue. In future we would take larger bladder specimens from the NHIC patients with bladder wall thickening for more detail histology investigation.
- The bladder wall of patients with end-stage HIC eventually get thinner. Thus I guess the manuscript title should be changed as "Possible Association Between Bladder Wall Morphological Changes on Computed Tomography and Bladder-Centered Interstitial Cystitis/Bladder Pain Syndrome"
Reply: Thanks for your comment. We had revised the manuscript title.
Reviewer 2 Report
Dear Editor and Author, In my opinion, whole paper does not raise any substantive doubts. It is written in clear and transparent scientific language. All abbreviations are clearly explained. The methodology, results and conclusions are detailed. As a clinician who works with patients with IC, I'm very happy to see studies like that, because they improve the diagnostics of conditions such as IC. I fully recommend accepting the paper for publishing in the Biomedicines. RegardsAuthor Response
Reviewer 2:
Dear Editor and Author, In my opinion, whole paper does not raise any substantive doubts. It is written in clear and transparent scientific language. All abbreviations are clearly explained. The methodology, results and conclusions are detailed. As a clinician who works with patients with IC, I'm very happy to see studies like that, because they improve the diagnostics of conditions such as IC. I fully recommend accepting the paper for publishing in the Biomedicines. Regards
Reply: Thanks for your review and opinion for this manuscript.